# Targeted Therapeutic Approach Based on Understanding of Aberrant Molecular Pathways Leading to Leukemic Proliferation in Patients with Acute Myeloid Leukemia

**DOI:** 10.3390/ijms22115789

**Published:** 2021-05-28

**Authors:** Moo-Kon Song, Byeong-Bae Park, Ji-Eun Uhm

**Affiliations:** 1Department of Hematology-Oncology, Hanyang University Hanmaeum Changwon Hospital, Changwon 51497, Korea; mksong9676@hanmail.net; 2Division of Hematology-Oncology, Department of Internal Medicine, Hanyang University College of Medicine, Hanyang University Seoul Hospital, Seoul 51139, Korea; jieunuhm@hanyang.ac.kr

**Keywords:** acute myeloid leukemia, gemtuzumab ozogamicin, BCL-2

## Abstract

Acute myeloid leukemia (AML) is a heterogenous hematopoietic neoplasm with various genetic abnormalities in myeloid stem cells leading to differentiation arrest and accumulation of leukemic cells in bone marrow (BM). The multiple genetic alterations identified in leukemic cells at diagnosis are the mainstay of World Health Organization classification for AML and have important prognostic implications. Recently, understanding of heterogeneous and complicated molecular abnormalities of the disease could lead to the development of novel targeted therapeutic agents. In the past years, gemtuzumab ozogamicin, BCL-2 inhibitors (venetovlax), IDH 1/2 inhibitors (ivosidenib and enasidenib) FLT3 inhibitors (midostaurin, gilteritinib, and enasidenib), and hedgehog signaling pathway inhibitors (gladegib) have received US Food and Drug Administration (FDA) approval for the treatment of AML. Especially, AML patients with elderly age and/or significant comorbidities are not currently suitable for intensive chemotherapy. Thus, novel therapeutic planning including the abovementioned target therapies could lead to improve clinical outcomes in the patients. In the review, we will present various important and frequent molecular abnormalities of AML and introduce the targeted agents of AML that received FDA approval based on the previous studies.

## 1. Introduction 

Acute myeloid leukemia (AML) is a group of hematological malignancies characterized by rapid and uncontrolled growth of immature white blood cells in the bone marrow (BM) [1]. The various molecular alterations identified in leukemic cells at diagnosis are the mainstay of the World Health Organization classification for AML and have important prognostic implications. Some subtypes are associated with a favorable prognosis with intensive chemotherapy. However, the clinical outcome of AML is generally poor, with a worldwide 5-year overall survival rate of just 28% [2]. The prognosis is especially unfavorable in elderly patients, who tend to be ineligible for intensive chemotherapy; the median survival time in such patients is less than 1 year [3]. In addition, since the cytarabine and idarubicin regimen was established as the standard induction chemotherapy for AML, it has remained unchanged [4]. The regimen is of limited therapeutic efficacy in many different genetic subtypes of AML. Thus, novel effective therapies are needed for such patients.

The revolution in understanding the genetic alterations of AML that has been driven by next-generation sequencing has resulted in numerous therapeutic options against potential driver mutations such as FMS-like tyrosine kinase three-internal tandem duplication (FLT3-ITD) and isocitrate dehydrogenase (IDH) mutations [5]. The 2017 European Leukemia Net (ELN) criteria provide useful information to determine the best therapeutic option between conventional and novel therapies. According to the criteria, AML patients are separated into favorable, intermediate, and adverse risk groups [6]. In the German-Australian AML Study Group, the prognostic impact of many mutations is characterized by the combined effect of concomitant molecular abnormalities [7]. NPM1 mutation is associated with a favorable prognosis in the absence or very low allele ratio of the FLT3-ITD mutation. However, tumor protein 53 (TP53) mutation is strongly associated with adverse prognosis and mainly occurs in secondary or therapy-related AML, mostly characterized by complex cytogenetics. 

The use of hypomethylation agents (HMAs) or low-dose cytarabine (LDAC) treatment options in patients unfit for intensive chemotherapy or stem cell transplantation (SCT) were recently shown to be modestly effective but not satisfactory [8,9]. Advancements in our understanding of the genetic basis of AML over the last decade have led to the rapid development of targeted therapies. Complicated genetic mutations in AML patients could reflect several biological diseases classified by cytogenetically and molecularly defined risk. In addition, a large amount of data about novel targeted therapies for AML have shown promising results, particularly in patients without alternative therapeutic options. 

In this review, we will discuss the available clinical data on novel approved targeted therapeutic options for AML based on clinical trials conducted since 2017 that could lead to US Food Drug Administration (FDA) approval.

## 2. Mutational Landscape in AML

Leukemogenesis is a multistage process leading to clonal proliferation of abnormal blood cells derived from transformed primitive hematopoietic stem cells [10]. AML is characterized by the accumulation of immature leukemic cells in the BM or peripheral blood at the expense of the normal production of terminally differentiated blood cells. AML occurs at all ages, but incidence rates are higher in the elderly, with a median age of diagnosis of ~70 years [11]. 

Due to the development of diagnostic technologies for cytogenetic analysis in AML, recent in-depth analysis of data from a wide cohort clarified that the number of coding mutations per AML exome or genome is lower than in most other human cancers [12,13]. The mutations lead to the deregulation of pathways such as DNA methylation-associated genes, spliceosome-complex genes, cohesion-complex genes, chromatin-modifying genes, and signaling genes [14,15]. At least one driver mutation in one of these genes is present in about 96% of AML patients. 

Several studies have identified various novel recurrent somatic disease alleles with biological, prognostic, and therapeutic relevance, such as mutations of the DNA methyltransferase 3A gene (DNMT3A) and IDH 1/2 [16,17,18,19]. Other findings also indicate that AML is a complex and dynamic neoplasm with multiple somatically acquired driver mutations and coexisting competing clones and that the disease evolves over time [20,21,22]. Together, these results imply that molecular analysis could be used to predict a patient’s prognosis and thus has an important role in AML patient management. 

The most frequent genetic mutations in AML are summarized in Table 1. Although NPM1 mutation is associated with a good prognosis in AML patients who receive standard therapy, DNMT3A or FLT3-ITD mutations could counteract the prognostic significance of NPM1 mutations [23]. However, FLT3-ITD with a low allelic ratio may not influence the clinical impact of NPM1 mutation. 

In addition, TP53 mutation is strongly indicative of adverse prognosis in AML and is mostly characterized by secondary AML patterns and complex cytogenetics [15]. In this review, we will discuss the most important molecular mutations, such as CD33, FLT3-ITD, BCL2, and IDH1/2 mutations and aberrant hedgehog signaling, and the associated targeted approaches in patients with AML, along with the US FDA approval status of each approach.

## 3. Anti-CD33 Directed Antibody

### Mechanism of Action

Gemtuzumab ozogamicin (GO) is a CD33-directed antibody–drug conjugate (ADC) composed of h67.6, a CD33-directed monoclonal antibody, covalently linked to the cytotoxic agent N-acetyl γ calicheamicin [24]. The efficacy of GO is associated with the ubiquitous nature of CD33 as a potent target for immunotherapeutic options for AML. CD33 inhibits cell signaling by recruiting SHP-1 and 2 upon phosphorylation of tyrosine residues located within the immune-receptor tyrosine-based inhibitory motif domain on the cytoplasmic tail of the protein [25]. CD33 is internalized when it engages with antibodies. Notably, the activity of GO is derived from internalization of the ADC after successful binding of the monoclonal antibody to the immunoglobulin (Ig) V domain of CD33 on the surface of leukemic cells (Figure 1) [26].

Calicheamicin is a potent antitumor antibiotic from *Micromonospora echinospora* that is responsible for the cytotoxic activity of GO [27]. Once the GO-CD33 complex is internalized, the acidic lysosomal interior hydrolyzes the disulfide bond connecting calicheamicin to the acid-labile linker of GO, releasing free calicheamicin into the cell [24]. After the GO-CD33 complex is internalized, which occurs rapidly, the complex is routed to the lysosomes of the cytoplasm. In the acidic environment of the lysosome, the butanoic acid linker is hydrolyzed, releasing the toxic moiety of GO. The calicheamicin derivative is reduced by glutathione into a highly reactive species, which induces simple and double-stranded DNA breaks, resulting in DNA damage [28]. Then, the downstream DNA repair pathway is activated through the ataxia-telangiectasia mutated (ATM)/ataxia-telangiectasia and Rad3-related (ATR) and DNA-dependent protein kinase pathways and ATM/ATR proteins phosphorylate CHK1/CHK2 proteins, leading to G2M cell cycle arrest. ATM/ATR are two leading proposed DNA damage response pathways that are activated as a result of these breaks, leading to apoptosis of leukemic cells [29,30,31]. 

## 4. Gemtuzumab Ozogamicin, Anti-CD33 Antibody

### 4.1. Clinical Data 

GO initially received accelerated FDA approval in 2000 based on phase II clinical trial data. The trial revealed a benefit of GO as a single agent in patients over the age of 60 with CD33+ AML at a dose of 9 mg/m^2^/day on days 1 and 14 [32]. The data showed an objective response rate (ORR) of 30% and a complete response (CR) rate of 16.2%. In the 2004 post-approval phase III trial SWOG S0106 study, patients were randomized to receive either standard induction with daunorubicin (60 mg/m^2^/day on days 1, 2, and 3) and cytarabine (100 mg/m^2^/day from days 1–7) (DA) or a GO-containing induction with lower doses of daunorubicin (45 mg/m^2^/day on days 1, 2, and 3), cytarabine (100 mg/m^2^ from days 1–7) and GO (6 mg/m^2^ on day 4; DA + GO) [33]. The addition of GO did not show a clinical benefit but was associated with an increased early mortality rate. Interestingly, DA combined with GO improved relapse-free survival (RFS) among patients in the favorable cytogenetic risk group (hazard ratio [HR]; 0.49; *p* = 0.043). 

In Table 2, the phase III MRC AML15 trial enrolled 1113 patients younger than 60 years of age, who were randomized to receive a lower dose (3 mg/m^2^) of GO in induction 1 and in consolidation, in addition to the standard or other experimental treatments [34,35]. The study had three different induction arms, including ADE, DA, and Ida/FLAG. Overall, the addition of GO was well tolerated without a substantial increase in toxicity. However, based on the original GO randomization scheme, the addition of GO was not associated with improved outcomes. The only patients who benefitted from GO therapy were those with favorable karyotypes. Meanwhile, the group with intermediate or high cytogenetic risk showed no significant survival benefits. 

In contrast, the ALFA-0701 simple single open-label phase III study investigated the efficacy of a fractionated dose of GO, 3 mg/m^2^, in addition to standard induction chemotherapy in newly diagnosed de novo AML patients [36]. The patients were randomized to receive standard DA therapy or DA combined with GO therapy. GO was administered at 3 mg/m^2^ on days 1, 4, and 7 during induction therapy. The rates of remission (81% vs. 75%, *p* = 0.25) and 3-year OS (38% vs. 36%, *p* = 0.18) were not significantly improved in the GO group, while 3-year event-free survival (EFS; 31 vs. 19%, *p* = 0.0026) and 2-year RFS (38 vs. 25% *p* = 0.006) significantly improved in the GO group. 

Moreover, a meta-analysis of 3325 patients from five randomized controlled trials, the MRC AML15, SWOG S0106, NCRI AML16, GOELAMS AML 2006 IR, and ALFA-0701 trials demonstrated that the addition of GO did not increase the portion of patients achieving CR/CRi but significantly reduced the risk of relapse and improved OS at 5 years [37]. In addition, the data showed that the low dose of GO, 3 mg/m^2^, was associated with fewer early deaths than the higher dose of 6 mg/m^2^, while the two were equally efficacious. 

The NCRI AML17 trial was designed to identify the optimal dose of GO. The study patients were randomized to receive GO on day 1 of induction treatment at a dose of either 3 or 6 mg/m^2^. The CR rates were similar across both groups (82 vs. 87% *p* = 0.0) [38]. The ORRs were also similar (89% vs. 86%, *p* = 0.17). Moreover, the data showed that RFS and OS did not differ between the two groups (*p* = 0.5; *p* = 0.3). However, the 30-day mortality (7% vs. 3% *p* = 0.02) and 60-day mortality rates (9% vs. 5%, *p* = 0.01) were significantly higher in the 6 mg/m^2^ group than the 3 mg/m^2^ group (*p* = 0.02). Moreover, the incidence of veno-occlusive disease was higher in the 6 mg/m^2^ group (5.6% vs. 0.5%; *p* < 0.0001). The results revealed that there was no advantage in using a single dose of 6 mg/m^2^ GO compared with 3 mg/m^2^ in combination with induction therapy. In this regard, the lower fractionated GO dosing plan seems to produce better response and survival rates in a combination setting with standard induction therapy in AML patients.

Finally, the EORTC-GIMEMA AML-19 study showed an increase in OS rate in AML patients treated with GO-containing therapy, compared to best supportive care (BSC) when patients with newly diagnosed CD33-positive AML were treated with GO monotherapy at 6 mg/m^2^ on day 1 and 3 mg/m^2^ on day 8 [39]. The patients were randomly assigned to the two groups (118 to the GO group and 119 to the BSC group). The median OS was 4.9 months (95% CI, 4.2–6.8 months) in the GO group and 3.6 months (95% CI, 2.6–4.2 months) in the BSC group (HR, 0.69; 95% CI, 0.53–0.90; *p* = 0.005). The 1-year OS rate was 24.3% in the GO group and 9.7% in the BSC group. The OS benefit conferred by GO was consistent across most subgroups, and was especially apparent in patients with high CD33 expression status, in those with favorable/intermediate cytogenetic risk profiles, and in women. Overall, CR was observed in 30 of 111 (27%) patients in the GO group.

### 4.2. BCL-2 Inhibitor

#### Mechanisms of Action

The function of the BCL-2 protein is to prevent cellular apoptosis. Thus, overexpression of BCL-2 is significantly associated with inappropriate apoptosis, increased tumor overgrowth, and diminished sensitivity to chemotherapy [40]. In normal cells, antiapoptotic proteins such as BCL-2 and MCL-1 prevent apoptosis by constraining effector proteins (BAX and BAK) as cell death mediators. However, when cells are no longer required or undergo significant stresses, such as genotoxic damage, apoptosis is stimulated by activation of BCL-2 homology domain 3 (BH3)-only proteins such as BIM, BID, BAD, PUMA, NOKA, BIK, BMF, and HRK. These BH3-only proteins bind to and inhibit BCL-2 and MCL-1. Once BCL-2 is targeted in this manner, BAX and BAK cannot be constrained and drive cell death by causing mitochondrial damage. Since the binding of BH3-only proteins to BCL-2 or MCL-1 has a catalytic role in cell apoptosis, several substances that potently mimic their activity were developed to inhibit the activity of prosurvival proteins pharmacologically. Currently, the most advanced BH3 mimetic compound is venetoclax. Similar to the native BH3-only protein, venetoclax binds to BCL-2 with tight affinity, thereby relieving the constraint on BAX/BAK activation and initiating apoptosis (Figure 1) [41,42]. 

### 4.3. Venetoclax, BCL-2 Inhibitor

#### Clinical Data 

Preclinical data demonstrated that AZA could reduce MCL-1 levels, mediating resistance to BCL-2 inhibitors. AZA primed AML cells for venetoclax-induced apoptosis via NOXA induction. Thus, AZA and venetoclax synergistically activated BAX and thus stimulated mitochondrial apoptosis in AML cells [43,44]. 

In Table 2, a large, multicenter, phase Ib dose-escalation and expansion study reported on the safety and efficacy of venetoclax with HMA in AML patients older than 65 years with treatment-naive AML who were ineligible for intensive chemotherapy [42]. During dose escalation, oral venetoclax was administered at 400, 800, or 1200 mg daily in combination with either decitabine (DEC) (20 mg/m^2^, days 1–5, intravenously [IV]) or azacytidine (AZA) (75 mg/m^2^, days 1–7, IV or subcutaneously [SC]). In the expansion, 400 or 800 mg venetoclax was given with either DEC or AZA. The median patient age was 74 years, and poor-risk cytogenetics was present in 49% of patients. In a median follow-up time of 8.9 months, the CR/CRi rates were 67% and did not differ between the AZA and DEC groups. Patients with poor-risk cytogenetics and those older than 75 years had CR/CRi rates of 60% and 65%, respectively. The median time to response was 1.2 cycles (months) and the MRD negativity rate among responders was 29%. With a median follow-up time of 15 months, the median duration of response (DOR) and OS were 13.1 and 17.5 months, respectively. Among patients with CR/CRi, median DOR was 11.3 months, and median OS was not reached. Although benefits were seen in all patients, outcomes differed between the molecular and cytogenetic subgroups. Accordingly, CR/CRi rates were higher in patients with NPM1 and IDH1/2 mutations (91 and 71%, respectively) and lower in patients with TP53 mutations and poor cytogenetics (47 and 60%, respectively). Median DOR was also longer in patients with NPM1 and IDH1/2 mutations (24.4 months) and shorter in those with FLT3 and TP53 mutations (7.2 months). 

Another phase III, multicenter, randomized, double-blind, placebo-controlled trial was conducted to evaluate the efficacy and safety of AZA plus venetoclax, compared to AZA plus placebo in 431 newly diagnosed AML patients who were unfit for standard induction therapy due to coexisting comorbidities and age greater than 75 years old [45]. The patients were treated with AZA (75 mg/m^2^ SC or IV on day 1–7, 28-day cycle) plus venetoclax (target dose, 400 mg) or matching placebo administered orally in 28-day cycles. The intention-to-treat population included 431 patients (286 in the AZA-venetoclax group and 145 in the AZA–placebo control group). The median age was 76 years in both groups (range, 49–91 years old). At a median follow-up of 20.5 months, the median OS was 14.7 months in the AZA–venetoclax group versus 9.6 months in the control group (hazard ratio for death, 0.66; 95% CI, 0.52–0.85; *p* < 0.001). The incidence of CR was also higher in the AZA–venetoclax group than in the control group (36.7% vs. 17.9%; *p* < 0.001). Moreover, the composite CR and DOR were higher in the AZA-venetoclax group than in the control group (composite CR, 66.4 vs. 28.3%, *p* < 0.001; DOR, 17.8 vs. 13.9 months). The incidence of any-grade important adverse events (44% in the AZA-venetoclax group vs. 35% in the control group) and >grade 3 thrombocytopenia (45 vs. 38%, respectively), neutropenia (42 vs. 28%), febrile neutropenia (42 vs. 19%) and any-grade infections (84 vs. 67%) were also investigated.

A recent phase Ib/II study also investigated the safety and preliminary efficacy of venetoclax combined with LDAC in AML patients older than 60 years old and unfit for intensive chemotherapy. In the data, venetoclax (600 mg/day) was orally administrated in 28-day cycles, and LDAC (20 mg/m^2^ per day, SC) was given on days 1 to 10 [41]. The median age was 74 years (range, 63–90 years). Overall, 29% of the patients had previously received HMA, 49% had secondary AML, and 32% had poor-risk cytogenetic features. The early mortality rate was 6%. Moreover, 54% of the patients achieved CR/CRi. The median OS was 10.1 months (95% CI, 5.7 to 14.2), and the median DOR was 8.1 months (95% CI, 5.3 to 14.9 months). Among patients without prior HMA exposure, CR/CRi was achieved in 62% of cases, median DOR was 14.8 months, and median OS was 13.5 months (95% CI, 7.0 to 18.4 months). The most common grade 3 or greater adverse events were febrile neutropenia (42%), thrombocytopenia (38%), and neutropenia (34%). In addition, patients with NPM1 or IDH1/2 mutations had better outcomes (CR/CRi rate, 89 and 72%, respectively), compared to those with TP53 or FLT3 mutations (30 and 44%, respectively). The data showed that venetoclax plus LDAC showed a significantly improved safety profile, producing rapid and durable remission in older adults with AML ineligible for intensive chemotherapy. 

In another international phase III randomized double-blind placebo-controlled trial, patients older than 18 years old with newly diagnosed AML ineligible for intensive chemotherapy were randomized 2:1 to receive venetoclax (*n* = 143) or placebo (*n* = 68) in 28-day cycles, plus LDAC on days 1 to 10 [46]. The median age was 76 years old (range, 36–93 years), 38% of patients had secondary AML and 20% had received prior HMA treatment. The planned data analysis showed that venetoclax plus LDAC led to a 25% reduction in the risk of death, compared to LDAC alone, but the difference was not statistically significant (HR, 0.75; 95% CI, 0.52–1.07; *p* = 0.11), although median OS was 7.2 vs. 4.1 months, respectively. The unplanned analysis showed that the venetoclax arm had significantly greater OS (8.4 months; HR, 0.70; 95% CI, 0.50–0.98; *p* = 0.04). The CR/CRi rates were 48 and 13% in the venetoclax plus LDAC group and LDAC alone group, respectively (*p* < 0.001). The reported adverse events greater than grade 3 (venetoclax vs. LDAC alone) were febrile neutropenia (32 vs. 29%), neutropenia (47 vs. 16%), and thrombocytopenia (45 vs. 37%). Venetoclax plus LDAC treatment was associated with significant improvements in response and OS, compared to LDAC alone, with a favorable safety profile. 

In the abovementioned data, the efficacy of venetoclax-based chemotherapy was associated with prognosis in patients with specific cytogenetic abnormalities. In particular, patients with NPM1 and IDH 1/2 mutations treated with venetoclax-based chemotherapy had favorable median survival duration. The data suggest the utility of harnessing molecular strategies for patient selection to optimize response to venetoclax-based chemotherapy, particularly for AML patients with treatment-naïve NPM1 or IDH1/2 mutations unfit for intensive chemotherapy. 

### 4.4. FLT3 Inhibitors

#### Mechanism of Action

FLT3 is a cytokine receptor that is exclusively expressed in hematopoietic cells and plays a role in normal hematopoietic cell proliferation and survival [47]. Two frequently encountered activating FLT3 mutations include internal tandem duplications (ITDs) in the juxtamembrane domain and point mutations in the tyrosine kinase domain (TKD), most commonly at codon D835 [48,49]. FLT3 mutations are significantly involved in the proliferation, differentiation, and apoptosis of hematopoietic cells. FLT3-ITD mutation is observed in about 25% of patients with AML, whereas FLT3-TKD is detected in 7–10% of patients [50].

The presence of ITD causes loss of this inhibitory effect, leading to activation of TKD. The loss of the inhibitory effect of FLT3 is independent of the size of the duplication within the receptor. In addition, ITD-induced FLT3 signaling is aberrant, notably involving activation of STAT5 and its downstream effector molecules including Pim-1 kinase [51]. Although both FLT3-ITD and FLT3-TKD mutations result in constitutive activation of FLT3 signaling, downstream signaling pathways differ between the two mutations [52]. FLT3-ITD stimulates FLT3 signaling through JAK2/STAT5, with PI3K/AKT/mTOR and RAS/MEK/ERK. A previous preclinical study showed that STAT5 positively regulates Pim-1, which eventually activates mTOR and MCL-1, consequently conferring resistance to AKT inhibition in the FLT3-ITD cell line [53]. In other recent data, FLT3 mutations were shown to activate STAT5, leading to Bcl-x and RAD-51 upregulation, which accounted for drug resistance. However, FLT3-TKD mutations activate AKT and ERK signaling but not STAT5 (Figure 1) [54]. 

AML cells with FLT3-ITD had a high degree of genetic instability due to both an increase in DNA double-strand breaks associated with increased generation of reactive oxygen species and error-prone DNA double-strand break repair. Thus, the mutation is associated with poor treatment outcomes and short relapse-free and overall survival rates [55]. Recently, targeting of the FLT3 mutation in AML has been investigated with numerous type I tyrosine kinase inhibitors that bind the gatekeeper domain and type II inhibitors that bind the activation loop. 

### 4.5. Midostaurin, FLT3 Inhibitor

#### Clinical Data

Midostaurin is a small molecule tyrosine kinase inhibitor (TKI) that promotes direct and indirect inhibition of mutant FLT3 receptor phosphorylation [56]. It has been shown to induce cell cycle arrest and apoptosis in both FLT3-ITD and FLT3-D835Y mutant cell lines. Thus, this agent was approved by the FDA in 2017 for the treatment of AML patients with FLT3-ITD mutation [57]. 

A phase Ib trial by Stone et al. investigated the efficacy of midostaurin combined with daunorubicin and cytarabine induction therapy [58]. The CR rate associated with twice-daily administration of midostaurin 50 mg was similar between the FLT3-ITD mutation group (92%) and the FLT3-WT group (74%). Moreover, the 1- and 2-year OS rates were similar in patients with FLT3-ITD mutation (0.85 and 0.62) and FLT3-WT (0.78 and 0.52).

In Table 2, a phase III RATIFY placebo-controlled study investigated use of induction and consolidation chemotherapy combined with midostaurin, followed by maintenance with midostaurin in 717 patients with newly diagnosed AML with FLT3 mutation [59]. Although there was no significant difference in CR rate between the two groups, patients treated with midostaurin achieved significantly longer EFS and OS (*p* = 0.009 and *p* = 0.002, respectively). The improved OS in patients with low (0.05–0.7) and high FLT3-ITD allelic burden given midostaurin suggests that the therapeutic mechanism of action may not be solely due to FLT3 kinase inhibition but may include inhibition of multiple kinases. The clinical benefit of midostaurin was consistent in patients with both FLT3-ITD and TKD mutations, and the rate of severe toxicity was similar in the two groups. These data led to FDA approval of midostaurin in newly diagnosed AML patients with FLT3 mutation. 

### 4.6. Quizartinib, FLT3 Inhibitor

#### Clinical Data

Quizartinib is a selective and potent TKI of FLT3-ITD mutation and FLT3-WT without activity on FLT3-TKD [60]. It is a selective small-molecule inhibitor as monotherapy in the R/R setting and has shown enriched responses in patients with FLT3-ITD mutations. Quizartinib is generally well tolerated, with important adverse effects greater than grade 3 including QTc prolongation, bone marrow suppression, fatigue, and hypoalbuminemia. 

In a phase I study of quizartinib in 76 patients with relapsed/refractory AML irrespective of FLT3 status, the ORR was 30%, including a 13% CR rate of any type [61]. Importantly, responses were higher in patients with FLT3-ITD mutation, compared with FLT3-WT (ORR, 53 vs. 14%, respectively). In patients with FLT3-intermediate/not tested status, 41% showed a response. The median response duration was 13.3 weeks. The key adverse effect of quizartinib was QTc prolongation, which occurred in 12% of patients. 

A subsequent similar phase II study investigated the efficacy of quizartinib in patients with R/R AML regardless of FLT3 status [62]. The study enrolled a total of 333 patients, of which 157 were in cohort 1, who were more than 60 years old and had R/R AML within 1 year after first-line therapy, and 176 were in cohort 2, who were over 18 years of age and had R/R disease following salvage chemotherapy or SCT. In cohort 1, 56% of FLT3-ITD-positive patients and 36% of FLT3-ITD-negative patients achieved composite CR, while 3% of FLT3-ITD-positive patients and 5% of FLT3-ITD-negative patients achieved CR. In cohort 2, 46% of FLT3-ITD-positive patients achieved composite CR, and 4% achieved CR, while 30% of FLT3-ITD-negative patients achieved composite CR, and 3% achieved CR. In total, 38% of the patients died within the study treatment period, including 5% who died due to adverse effects. 

The recent QuANTUM-R trial aimed to compare quizartinib monotherapy with investigator’s choice such as low dose cytarabine, MEC, or a FLAG-Ida regimen in R/R AML patients with FLT3-ITD mutation [63]. The primary endpoint was OS in the intention-to-treat population. The results showed that OS was superior in the quizartinib group (median, 6.2 months), compared with the chemotherapy group. The rate of treatment-emergent deaths due to adverse events in the quizartinib group (13%) was comparable to that in the chemotherapy group (17%). Research in a front-line setting is ongoing, in the form of a phase III trial for patients with FLT3-ITD-positive AML. However, the development of FLT3 point mutations such as the D835 mutation is associated with acquired resistance to quizartinib. 

### 4.7. Gilteritinib, FLT3 Inhibitor

#### Clinical Data

Gilteritinib is a dual selective inhibitor of FLT3 and AXL [64]. It shows potential efficacy against FLT3-ITD and D835 mutations and concurrently inhibits AXL kinase, which is associated with FLT3 inhibitor resistance. TKI is approved by the FDA as a single agent for R/R AML. 

In Table 2, /II study of 252 R/R AML patients, gilteritinib was well tolerated, with 37% of patients with FLT3-ITD mutation achieving composite CR, along with 9% of patients with FLT3-WT [65]. Of the patients who received prior sorafenib therapy, 49% achieved composite CR. These findings demonstrate that gilteritinib may be able to overcome some of the acquired resistance mechanisms observed in response to preceding FLT3-TKI treatment. 

Moreover, a recent phase III ADMIRAL trial randomly assigned 371 patients with relapsed or refractory FLT3-mutated AML to receive either gilteritinib 120 mg/day or salvage chemotherapy [66]. The median OS in the gilteritinib group was significantly longer than that in the chemotherapy group (9.3 vs. 5.6 months; *p* < 0.001). The median EFS was not significantly different between the two groups (2.8 months in the gilteritinib group vs. 0.7 months in the chemotherapy group). The percentage of patients who achieved CR with full or partial hematologic recovery was 34.0% in the gilteritinib group and 15.3% in the chemotherapy group (risk difference, 18.6 percentage points; 95% CI, 9.8 to 27.4). The ORR was higher in the gilteritinib group than the chemotherapy group. In addition, adverse events greater than grade 3 occurred less frequently in the gilteritinib group than in the chemotherapy group; the most common events in the gilteritinib group were febrile neutropenia (45.9%), anemia (40.7%), and thrombocytopenia (22.8%).

### 4.8. Isocitrate Dehydrogenase Inhibitors

#### Mechanism of Action

The isocitrate dehydrogenase (IDH) enzyme promotes the turnover of isocitrate to α-ketoglutarate (α-KG) under normal conditions. IDH2 is localized in the mitochondria and operates within the citric acid cycle, while IDH1 promotes the same reaction within the cytoplasm [67]. 

IDH is a key enzyme in the Krebs cycle that catalyzes the oxidative decarboxylation of isocitrate to α-ketoglutarate. Mutations in the catalytic domains of IDH1/2 result in the reduction of α-ketoglutarate to R2-hydroxyglutarate (R2-HG) as an oncometabolite [68,69]. R2-HG competitively inhibits α-ketoglutarate-dependent enzymes, leading to DNA and histone hypermethylation, chromatin modification, and differentiation arrest of hematopoietic cells (Figure 1).

IDH mutations occur in about 20% of AML cases (5 to 13% IDH1 and 8 to 17% IDH2 mutations) [70,71,72,73]. IDH mutations are more frequently identified in older patients, intermediate-risk patients, and those with higher platelet counts, increased BM counts, or with FLT3-ITD and NPM1 mutations [14,74]. The frequency of IDH mutations increases to 10–20% at the time of leukemic transformation and are often seen with DNMT3A, ASXL1, and SRSF2 co-mutations [75]. Allosteric IDH inhibitors such as enasidenib and ivosidenib effectively suppress the production of R2-HG, leading to a reduction in the proliferation of leukemic cells [76,77].

### 4.9. Enasidenib, IDH2 Inhibitor

#### Clinical Data 

Enasidenib is a bivalent inhibitor of R140Q and R172K mutated IDH2 and is the first IDH mutation-specific inhibitor [78,79]. It promotes the terminal differentiation of myeloid blasts into neutrophils in vivo. 

A phase 1/2 study evaluated enasidenib doses of 50 to 650 mg/d in 239 patients with mutant-IDH2 AML [80]. In the study, the median age was 68 years old, ORR was 38.8% (95% CL 32.2–45.7), and CR rate was 19.6%. Median OS was 8.8 months (95% CI, 7.7–9.6) and response and survival were comparable between the IDH2-R140 group and IDH2-R172 mutation group. Red blood cell transfusion-independence was achieved in 43.1% of patients, and platelet transfusion-independence was achieved in 40.3%. The magnitude of 2-HG reduction was associated with CR in IDH2-R172 patients. Clearance of mutant-IDH2 clones was also associated with the achievement of CR. These data suggest that molecular remission is correlated with CR. The most common grade 3 or 4 treatment-related adverse events were hyperbilirubinemia (10%), thrombocytopenia (7%), and IDH differentiation syndrome (6%). Enasidenib was well tolerated and induced molecular remission and hematologic responses in patients with AML for whom prior treatments had failed. In the molecular analysis, the response was associated with a reduction in IDH2 allele burden and molecular clearance. However, clearance of specific co-mutations, such as KRAS, NRAS, or FLT3 mutations, was associated with a poor response rate. FDA approval of enasidenib was granted based on the above data in R/R AML patients with IDH2 mutations. 

Another recent retrospective study investigated real-world outcomes among a large cohort of patients with R/R AML with IDH2 mutations treated with enasidenib or other systemic therapies in the first R/R setting [81]. Of the 124 patients in the enasidenib group and 76 in the control group, 52% and 55% were male (*p* = 0.62), and the median age in the R/R setting was 68 and 63 years (*p* = 0.04), respectively (Table). The proportion of patients with European Cooperative Oncology Group performance status (ECOG PS) ≥2 (52% vs. 53%) and poor cytogenetic risk (29% vs. 29%) were similar between the two groups. Approximately 23% of patients in the enasidenib group were refractory to induction therapy, versus 40% in the control group (*p* = 0.02). The CR/PR/morphologic LFS rate was higher among patients treated with enasidenib than the control group (77% vs. 52%; *p* < 0.01). After a median follow-up duration of 9 and 6 months in the enasidenib group and control group, median PFS was 8.4 and 4.8 months (adjusted HR = 0.36, 95% CI 0.23–0.57; *p* < 0.01), and median OS was 11.0 and 6.4 months (adjusted HR = 0.37, 95% CI 0.22–0.60; *p* < 0.01), respectively. The results showed that superior response rate, PFS, and OS were observed in the enasidenib group, compared with the control group treated with other therapies. 

In addition, a retrospective single-center study on the use of enasidenib in R/R AML patients with IDH2 mutations analyzed outcomes in nine IDH2-mutated patients: four (44%) cases of de novo AML and five (56%) of secondary AML. The control group consisted of 28 patients [82]. The median age at relapse was 71 years old (range, 47–79 years). Median OS in the enasidenib group was 28 months from diagnosis (range, 3–65 months), and 15 months from treatment (range 1–27). Median PFS was 13 months (range 1–14). Among the 28 patients in the control group, median OS was 14 months (range 7–62 months) and OS from the last relapse was 2 months (range 0.5–26 months). The ENA patient group showed a significantly better OS than the control population (*p* = 0.0419). The data also demonstrated that the drug is generally well tolerated. 

The addition of enasidenib to induction, consolidation, and maintenance therapy for patients with newly diagnosed AML patients with IDH2 mutations is currently being evaluated in the randomized phase 3 trials. 

### 4.10. Ivosidenib, IDH1 Inhibitor

#### Clinical Data 

Ivosidenib is a potent and selective IDH1 mutation inhibitor that has shown promising results in phase 1 trials [83]. Overall, 258 patients received ivosidenib, and safety outcomes were assessed. There were 125 R/R patients in the primary efficacy population, and the median follow-up duration was 14.8 months. The rate of CR or CR with partial hematologic recovery was 30.4% (95% CI, 22.5–39.3), the CR rate was 21.6% (95% CI, 14.7–29.8), and the ORR was 41.6% (95% CI, 32.9–50.8). The median duration of response was 8.2, 9.3, and 6.5 months, respectively. After medication follow-up for a median of 14.8 months, the median OS was 8.8 months. Transfusion independence was attained in 35% of cases. Importantly, 88% of patients who achieved CR showed MRD negativity and 41% showed IDH1 mutation clearance. Thus, ivosidenib achieved reliable response and MRD negativity in older high-risk AML patients with IDH1 mutations. This trial led to FDA approval of ivosidenib in R/R AML patients with IDH1 mutations. 

Another study compared clinical outcomes in the ivosidenib group and historical control group in R/R AML patients with IDH1 mutations [84]. In total, 109 patients in the ivosidenib group were compared to 60 in the control group. Median OS was 8.1 months in the ivosidenib group compared with 2.9 months in the control group (*p* < 0.0001). The 6- and 12-month survival rates in the ivosidenib group were 57.7% (95% CI: 48.2, 67.2) and 35.0% (95% CI: 25.7, 44.3), respectively, while the survival rates in the control group were 29.1% (95% CI: 17.4, 40.8) and 10.8% (95% CI: 2.7, 18.9). The CR rate was also higher in the ivosidenib group (18.3%, 95% CI: 11.6, 26.9), compared to the control group (7.0%, 95% CI: 1.5, 19.1). Ivosidenib monotherapy was associated with prolonged OS and the potential to increase CR rates vs. standard of care therapies in the control group.

The clinical benefit of ivosidenib combined with other agents is currently being evaluated in the randomized phase 3 trials.

## 5. Hedgehog Signaling Pathway

### 5.1. Mechanism of Action

The hedgehog (HH)/glioma-associated oncogene homolog (GLI) signaling pathway is essential for embryonic development and is usually silenced in adult tissue [85]. Aberrant activation of HH/GLI signaling may play a major role in several cancers since it leads to genomic instability, proliferative signaling, replicative immortality, tumor invasion-metastasis, inflammation, and immune-surveillance evasion [86,87,88,89,90]. Aberrant activation of the HH/GLI pathway may result from both ligand-dependent and -independent mechanisms. In ligand-dependent activation, the so-called canonical HH/GLI pathway directly stimulates malignant cells [86]. The binding of processed and posttranslationally modified HH protein to its receptor, PTCH, abolishes the inhibitory effect of PTCH on SMO, allowing ciliary transport and activation of SMO. GLI zinc finger transcription factors (GLI1, GLI2, and GLI3) are activated by SMO activation, and activated GLI is translocated into the nucleus and binds to the target DNA of the promoter, leading to the expression of specific genes, such as those encoding c-MYC, BCL-2, and SNAIL. Suppressor of fused (SUFU) is one element in the canonical HH pathway that downregulates GLI1-mediated target genes. When the GLI activator is promoted by SMO, it is translocated to the nucleus, leading to induction of HH target gene expression. 

Meanwhile, ligand-independent HH/GLI signaling activation is mediated by PTCH loss-of-function or SUFU protein mutations, gain-of-function SMO mutations, or SMO-independent GLI activation through PI3K/AKT and RAS/RAF/MEK/ERK signaling [91]. Ligand-independent activation is considered the noncanonical means of HH pathway activation. Gene amplification or frameshift mutations finally result in abnormal GLI expression and function [92]. 

Overall, the HH/GLI signaling pathway is essential to hematopoietic stem cell function. Both HH and the signal transducer SMO are expressed in CD34+ AML cells. The HH/GLI signaling pathway is significantly associated with the resistance of AML cells to standard chemotherapy (Figure 1).

### 5.2. Glasdegib, HH/GLI Signal Pathway Inhibitor

#### Clinical Data

Glasdegib is a potent and selective oral inhibitor of the HH/GLI signaling pathway that acts by binding to SMO [93]. In preclinical studies, glasdegib, as a single agent or in combination with chemotherapy, was shown to reduce the expression of key leukemia stem cell regulators and decrease leukemia stem cell populations in patient-derived AML cells [93,94]. 

In Table 2, a randomized, open-label, multicenter study evaluated the efficacy of glasdegib plus LDAC in patients with AML or high-risk MDS who were unfit for intensive chemotherapy [95]. Glasdegib 100 mg was administered continuously in 28-day cycles and LDAC 20 mg SC twice/day was administered for 10 per 28 days. The patients, who were stratified by cytogenetic risk, were randomized (2:1) to receive glasdegib plus LDAC (*n* = 88) or LDAC (*n* = 44). Median OS was 8.8 months (range, 6.9–9.9 months) in the glasdegib plus LDAC group and 4.9 months (range, 3.5–6.0 months) in the LDAC group (HR, 0.51; 80% CI, 0.39–0.67, *p* = 0.0004). The CR rate was 17% in the gladegib plus LDAC group (*n* = 17), compared to 2.3% (*n* = 1) in the LDAC group (*p* < 0.05). Nonhematologic grade 3 or 4 adverse events included pneumonia (16.7%) and fatigue (14.3%) in the glasdegib plus LDAC group and pneumonia (14.6%) in the LDAC group. Clinical efficacy was evident across patients with diverse mutational profiles. Thus, the data showed that glasdegib plus LDAC has a favorable benefit–risk profile and may be a promising option for AML patients unfit for intensive chemotherapy.

The above clinical phase II trial led to FDA approval of the agent as a therapeutic option in newly diagnosed AML patients who are older than 75 years of age or who have comorbidities that make them unsuitable for intensive chemotherapy. 

## 6. Conclusions 

The standard treatment for AML includes intensive chemotherapy, followed by post-remission consolidation therapy with HSCT or chemotherapy. However, the modern management of AML has been significantly improved by the availability of novel targeted agents, such as an anti-CD33 monoclonal antibody, a BCL-2 inhibitor (venetoclax), FLT3 inhibitors (midostaurin, quizartinib, and gilteritinib), and IDH1/2 inhibitors (ivosidenib and enasidenib). Thus, as our understanding of the molecular biology of AML improves, it may be possible to predict targetable and novel prognostic subpopulations. 

GO should be considered in cases of newly diagnosed or R/R CD33+ AML with favorable or intermediate-risk cytogenetics. GO could improve OS when added to intensive chemotherapy. Screenings for FLT3 and IDH1/2 mutations are recommended at both the diagnosis and relapse stages because patients with any of these mutations will benefit from the incorporation of targeted FLT3 inhibitors and IDH1/2 inhibitors. Particularly in elderly patients unfit for intensive chemotherapy, combination therapy with venetoclax and HMA/LDAC or the HH/GLI signaling pathway inhibitor glasdegib is a potential therapeutic option.

Currently, the application of NGS at diagnosis to investigate AML-associated mutations is increasingly recognized as means of refining the clinical outcome of the patients, identifying biomarkers of therapeutic response, and selecting patients who may benefit from novel targeted therapies. 

Indeed, the increased diversity of therapeutic options requires a distinctive treatment algorithm that incorporates mutation-specific targeted therapies, monoclonal antibodies, and apoptosis-inducing small molecules. In this review, we anticipate that the use of a novel approved targeted agent could lead to improvement in clinical outcomes in patients with AML. Further well-designed clinical studies could address the mechanisms and clinical evidence of efficacy and safety of the targeted agents in patients with special conditions. 

## Figures and Tables

**Figure 1 ijms-22-05789-f001:**
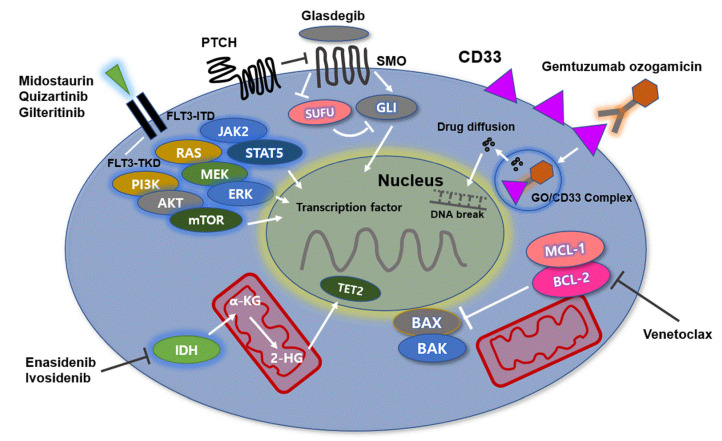
**The molecular mechanisms of AML:** Molecular dysregulation alters the expression profile of genes such as CD33, IDH1/2, FLT3, and BCL-2. The activity of gemtuzumab ozogamicin (GO) is derived from internalization of the CD33-GO complex after successful binding of the monoclonal antibody on the surface of leukemic cells, leading to apoptosis of leukemic cells. FLT3 mutations stimulate downstream signaling through JAK2/STAT5, PI3K/AKT/mTOR, and RAS/MEK/ERK. AML cells with FLT3-ITD mutations have a high genetic instability due to DNA double-strand breaks and are associated with poor clinical outcomes. Midostaurin, quizartinib, and gilteritinib effectively inhibit FLT3-ITD mutations. IDH1/2 mutation leads to reduction of α-ketoglutarate to R2-hydroxyglutarate (R2-HG) as an oncometabolite. IDH inhibitors inhibit production of R2-HG and thus block proliferation of leukemic cells. BCL-2 and MCL-1 prevent apoptosis of leukemic cells by regulating effector proteins such as BAX and BAK as cell death mediators. The native BH3-only protein venetoclax binds to BCL-2, thereby relieving the constraints on BAX/BAK activation and initiating apoptosis. The HH/GLI signaling pathway is associated with hematopoietic stem cell function. In leukemia cells, the signaling pathway is involved in resistance of AML cells to chemotherapy. Glasdegib effectively inhibits the HH/GLI signaling pathway by binding to SMO. **Abbreviations**: IDH, isocitrate dehydrogenase; FLT3, FMS-like tyrosine kinase 3; BCL-2, B-cell lymphoma-2; JAK2/STAT5, Janus kinase 2/signal transducer and activator of transcription 5; PI3K/AKT/mTOR, phosphoinositide 3-kinase/Akt/mechanistic target of rapamycin; RAS/MEK/ERK, rat sarcoma/rapidly accelerated fibrosarcoma/extracellular signal-regulated kinase; ITD, internal tandem duplication; HH/GLI, hedgehog/glioma-associated oncogene homolog; AML, acute myeloid leukemia.

**Table 1 ijms-22-05789-t001:** Mutational landscape in acute myeloid leukemia.

Mutation	Frequency of CN-AML	Mechanism of Action	Clinical Impact	Agents
NPM1	30–43%	Nucleolar component	Favorable	NA
DNMT3A	34%	De novo DNA methylation	ND	NA
FLT3-ITD	28–34%	FLT3 receptor tyrosine kinase	Unfavorable in high ratio (≥0.5)	Midostaurin, Sorafenib, Quizartinib, Gilteritinib
FLT3-TKD	11–14%	FLT3 receptor tyrosine kinase	Neutral	Quizartinib, Gilteritinib
IDH1/2	15–30%	Conversion of isocitrate to α-ketoglutarate	Favorable	Ivosidenib, Enasidenib
TET2	10%	Conversion of 5-methylcytosine to 5-hydroxy-methylcytosine (methylation)	ND	NA
ASXL1	5–16%	Epigenetic regulation by interaction with PRC2	Unfavorable	NA
CEBPA	10–18%	Hematopoietic transcription factor	Favorable	NA
RAS	25% NRAS, 15% KRAS	G-protein associated with receptor tyrosine kinase	Neutral	Cabimetinib
KIT	20–30% of CBF-AML	Receptor tyrosine kinase for stem cell factor	Unfavorable	Dasatinib, Imatinib
KMT2A-PTD	5–10%	abrogation of KMT2A transactivation and histone methyltransferase function	Unfavorable	NA
RUNX1	5–13%	Hematopoietic transcription factor	Unfavorable	NA
TP53	5–20%	Tumor-suppressor gene	Unfavorable	Idasanutlin

**Abbreviations**: NPM1, nucleophosmin 1; DNMT3A, DNA methyltransferase 3 alpha; FLT3-ITD, FMS-like tyrosine kinase 3-internal tandem duplication; IDH, isocitrate dehydrogenase; TET2, Tet methylcytosine dioxygenase 2; ASXL1, ASXL transcriptional regulator 1; CEBPA, CCAAT enhancer binding protein alpha; RAS, rat sarcoma; KMT2A, K-specific methyltransferae 2A; RUNX1, RUNXX family transcription factor 1; TP53, tumor protein 53; ND, not defined; NA, no agent.

**Table 2 ijms-22-05789-t002:** Clinical trials on novel targeted therapies for acute myeloid leukemia patients.

Author (Refer.)	Therapeutic Schedule	Phase/Population	Clinical Outcome
**Anti-CD33 monoclonal antibody**			
Petersdorf et al.	GO—6 mg/m^2^ on day 4. additional 3 doses, 5 mg/m^2^ in CR patients after consolidation GO + modified DA vs. standard DA	Phase III, ND AML, *n*= 595	ORR, 76% in DA plus GO group vs. 74% in DA alone (*p* = 0.36) CR, 69% vs. 70% (*p* = 0.69) 5-yr RFS, 47% vs. 42% (*p* = 0.87) 5-yr OS, 46% vs. 50% (*p* = 0.09)
Castaigne et al. ALFA-0701	DA +/− GO—3 mg/m^2^ for day 1, 4, and 7 of induction, 3 mg/m^2^ for day 1 of two consolidations	Phase III, ND AML, *n* = 278	CR/CRi, 81 in GO + group vs. 75% in GO—group (*p* = 0.25) 2-yr EFS, 40.8 vs. 17.1% (*p* = 0.0003) 2-yr OS, 53.2 vs. 41.9% (*p* = 0.0368) 2-yr RFS, 50.3 vs. 22.7% (*p* = 0.0003) Survival benefit—favorable and intermediate-risk group
Burnett et al. MRC-AML15	GO—3 mg/m^2^ for day 1 + DA, 2 cycles, FLAG-ida or ADE	Phase III, ND AML, *n* = 1113	Addition of GO—no different in OS, RFS, and TRM. But, OS ↑ in favorable cytogenetic risk (79 vs. 51%, *p* = 0.0003)
Burnett et al. NCRI-AML16 and LRF AML 14	GO—3 mg/m^2^ for day 1 + DA or DC (daunorubicin + claforabine, D 1-5)	Phase III, ND AML, *n* = 1115	IC—↑ RFS (28 vs. 23%, *p* = 0.03) and ↑ CR (35 vs. 29 and, *p* = 0.04) Non-IC—↑ ORR (17 vs. 30%, *p* = 0.006) and ↑ CR (11 vs. 21%, *p* = 0.002) But, no improvement of OS
Burnett et al. NCRI-AML17	GO—3 mg or 6 mg/m^2^ for day 1 + DA or ADE (DA + etoposide)	Phase III, ND AML, *n* = 788	Significant higher CR rate in 3 mg GO group vs. 6 mg group (*p* = 0.03) 6 mg group—higher 30 and 60-day TRM (*p* = 0.02; *p* = 0.01)
Delaunay et al. GEOLAMS-AML 2006 IR	GO—6 mg/m^2^ for day 1 + DA	Phase III, ND AML, *n* = 238	CR—not different between GO + vs. GO- group (91.6 vs. 86.5%, *p* = NS) EFS, OS—not different between GO + vs. GO- group.VOD, hepatotoxicity, higher in GO + group (23 vs. 13%; *p* = 0.031)
Burnett et al. EORTC-GIMEMA AML 19	GO—6 mg/m^2^ for day 1, 3 mg/m^2^ for day 8 vs. Best supportive care	Phase III, ND AML unfit for IC, *n* = 237	OS, 4.9 months in GO group vs. 3.6 months BSC group (*p* = 0.005) 1-yr OS, 24.3% vs. 9.7% OS benefit of GO, higher in women and favorable, intermediate-risk group. CR + CRi in GO group, 27%
**BCL-2 inhibitor**			
**Combination study with hypomethylating agents**			
DiNardo et al. Blood 2019	Venetoclax, 400, 800, 1200 mg + HMAs (AZA, or DEC)	ND AML ≥ 60 years or unfit for IC, *n* = 145	CR/CRi, 67% in all patients; CR/CRi, 73% in venetoclax 400 mg/day group Median CR/CRi duration, 11.3 months Median OS, 17.5 months
DiNardo et al. NEJM 2020	Venetoclax, 400 mg/day + AZA	Phase III, ≥75 years or unfit for IC, *n* = 431	OS, 14.7 months in venetoclax-AZA group vs. 9.6 months in control (*p* < 0.001) CR/CRi, 36.7%/66.4% in venetoclax-AZA group vs. 17.6%/28.3% in control (*p* < 0.001)
**Combination study with Low dose cytarabine**			
Wei et al. (JCO)	Venetoclax, 600 mg/day + LDAC		Median age, 74 yrs (range, 63–90 yrs) In enrolled patients CR/CRi, 54%; OS, 10.1 months; DOR, 8.1 months In patients without prior HMA exposure, CR/CRi, 62%; DOR, 14.8 months; OS, 13.5 months
Wei et al. (blood)	Venetoclax, from 100 mg/day to 600 mg/day + LDAC	ND AML unfit for IC, *n* = 211	Median age, 76 yrs (range, 36–93 yrs) OS, 8.4 mos in venetoclax + LDAC vs. 4.1 mos in LDAC alone (*p* = 0.04). CR/CRi, 48% in venetoclax + LDAC vs. 13% in LDAC alone (*p* < 0.001)
**FLT3 inhibitor**			
**Midostaurin**			
Stone et al.	Midostaurin, 50 mg/day twice/day + DA	Phase Ib, ND AML, *n* = 29	CR, 92% in FLT3-ITD + vs. 74% in FLT3-WT (*p* = NS) 1 and 2-yr OS, 0.85, 0.62 in FLT3-ITD+ vs. 0.78, 0.52 in FLT3-WT (*p* = NS) 1-yr DFS, 50 in FLT3-ITD+ vs. 60% in FLT3-WT (*p* = NS)
Stone et al.	DA +/− Midostaurin, 50 mg/day twice/day	Phase III, ND AML, *n* = 717	OS, 74.7 in midostaurin, higher than 25.6 months in placebo (*p* = 0.009) EFS, in midostaurin group, higher than placebo (*p* = 0.002) CR, 58.9 in midostaurin vs. 53.5% in placebo (*p* = NS). Midostaurin, beneficial in both ITD and TKD mutation Severe toxicity, similar between two groups (*p* = NS)
**Quizartinib**			
Cortes et al. (JCO)	quizartinib, escalating doses of 12 to 450 mg/day	Phase I, R/R AML +/− FLT3 status, *n* =76	In enrolled patients—ORR/CR—30%/13% ORR—53% in FLT3-ITD group vs. 14% FLT3-WT group
Cortes et al. (lancet)	quizartinib monotherapy	Phase II cohort, R/R AML, *n* = 333 Cohort 1 ≥ 60 yrs, R/R within 1 yr Cohort 2 ≥ 18 yrs, R/R after salvage or SCT	Cohort 1 Composite CR/CR—56%/3% in FLT3-ITD group Compositive CR/CR—36%/5% in FLT3-WT group Cohort 2 Composite CR/CR—46%/4% in FLT3-ITD group Compositive CR/CR—30%/3% in FLT3-WT group
Cortes et al.	quizartinib vs. investigator’s choice	Phase III, R/R AML with FLT-ITD +, *n* = 367	OS, 6.2 in quizartinib vs. 4.7 months in chemotherapy (*p* = 0.02) Therapy-related death, 17% vs. 17% (*p* = NS)
**Gilteritinib**			
	gilteritinib, 120 mg/day vs. salvage chemotherapy	Phase III, R/R AML with FLT-ITD +, *n* = 371	OS, 9.3 in gilteritinib vs. 5.6 months in chemotherapy (*p* < 0.001) EFS, 2.8 months vs. 0.7 months (*p* = NS). CR with hematologic recovery, 34.0 vs. 15.3% (18.6, 95% CI; 9.8-27.4)
**IDH1/2 inhibitor**			
**Enasidenib**			
Stein et al.	Dose-escalation phase, 50–650 mg/day/day Expansion phase, 100 mg/day.day	Phase I/2, R/R AML, *n*= 214	Median age, 68 years. ORR/CR—38.8%/19.6% BMT proceeding rate—10.3% Medians OS, 8.8 months RBC/PLT transfusion independence—40.2%/43.1%
Klink et al.	Enasienib, 50–650 mg/day/day Control group—other treatment group	Retrospective, R/R AML, *n* = 200	Enasidenib, less refractory to induction than control group (*p* = 0.02) CR/PR/LFS, enasidenib group, higher than control (*p* < 0.01) Median PFS, 8.4 vs. 4.8 months (*p* = <0.01) Median OS, 11.0 vs. 6.4 months (*p* < 0.01)
Riva et al.	Enasidenib, 100 mg/day/day Control group—other treatment group	Retrospective, R/R AML *n* = 37	Median OS in enasidenib, higher than control (*p* = 0.0419) PFS (*p* = NS)
**Ivosidenib**			
DiNardo et al.	ivosidenib 500 mg/d	Phase I, R/R AML, *n* = 125	Median follow-up duration, 14.8 monthsORR/CRh/CR—41, 30, 22% Duration of ORR/CRh/CR—6.5/8.2/9.3 months In F/U 14.8 months, median OS 8,8 months
Paschka et al.	Ivosidenib, 500 mg/day Control group—other treatment group	Data analysis, R/R AML, *n* = 434	OS, 8.1 in ivosidenib vs. 2.9 months control group (*p* < 0.0001) 6/12-month survival rate—55.7%/35.0 vs. 29.1%/10.8% (*p* < 0.001) CR—18.3% vs. 7.0% (*p* < 0.001)
**Hedgehog signaling inhibitor**			
**Glasdegib**			
Cortes et al.	Glasdegib, 100 mg + LDAC vs. LDAC alone	Phase II, ND AML unfit for IC, *n* = 132	Median OS was 8.8 months with glasdegib group vs. 4.9 months with LDAC group (*p* = 0.0004) CR, 17% in gladegib group vs. 2.3% in LDAC group (*p* < 0.05) Grade ≥ 3 AE, pneumonia (16.7%), fatigue (14.3%)

**Abbreviations**: GO, gemtuzimab ozogamicin; CR, complete response; ND AML, newly diagnosed acute myeloid leukemia; ORR, overall response rate; DA, daunorubicin and anthracycline; RFS, relapse-free survival; OS, overall survival; IC, intensive chemotherapy; VOD, veno-occlusive disease; CRi, morphologic complete remission with incomplete blood count recovery; LDAC, low dose cytarabine; DOR, duration of response.

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
