# Peer review of "Targeted Therapeutic Approach Based on Understanding of Aberrant Molecular Pathways Leading to Leukemic Proliferation in Patients with Acute Myeloid Leukemia"

_ijms, 2021, doi:10.3390/ijms22115789_

Round 1

Reviewer 1 Report

Song et al. have provided a comprehensive review on various targeted therapies for AML. The authors cover the subject matter well. However, the review could be made more navigable and attractive for readers before it is accepted for publication. Hence, I would suggest the following changes.

  1. Insert a summarizing table
  2. Add figures in the manuscript as appropriate to illustrate the various mechanisms of action

Author Response

Song et al. have provided a comprehensive review on various targeted therapies for AML. The authors cover the subject matter well. However, the review could be made more navigable and attractive for readers before it is accepted for publication. Hence, I would suggest the following changes.

  1. Insert a summarizing table

(answer) Table 2 is summarizing table. Please check the table.

  1. Add figures in the manuscript as appropriate to illustrate the various mechanisms of action

(answer) Figure 1 could include explain several mechanism as schematic figure. Please check the figure.

Reviewer 2 Report

If the presentation is suitably revised this review can provide an easy access to the recent Clinical Trials of targeted therapy for AML The following should be addressed.

  1. Better organization of the narrative. The main problem is that on lines 140 and 201 we see the name of an agent, not the target, while the following sub-sections, eg lines 272 , we do see the target, ie FLT3. Over all organization is also poor , as perhaps a heading for text on lines 140- 461 should also be provided.
  2. The sentence structure  should be reviewed. Eg.  The sentence on lines 66-67 is incomplete. Also, the sentence lines 487-489 seems to  imply the combination of conventional and targeted therapy? If so, it should be clarified. If not, what does it mean?
  3. Table 2 headings should also be modified as per comment #1
  4. A comprehensive list of abbreviations would improve clarity.

Author Response

If the presentation is suitably revised this review can provide an easy access to the recent Clinical Trials of targeted therapy for AML The following should be addressed.

  1. Better organization of the narrative. The main problem is that on lines 140 and 201 we see the name of an agent, not the target, while the following sub-sections, eg lines 272 , we do see the target, ie FLT3. Over all organization is also poor , as perhaps a heading for text on lines 140- 461 should also be provided.

(Answer) the healing problem is revised. Please check the words.

  1. The sentence structure should be reviewed. Eg.  The sentence on lines 66-67 is incomplete. Also, the sentence lines 487-489 seems to imply the combination of conventional and targeted therapy? If so, it should be clarified. If not, what does it mean?

(Answer) indicated sentences were revised. Please check the sentences.

  1. Table 2 headings should also be modified as per comment #1

(Answer) healings on Table 2 were also revised. Thank you.

  1. A comprehensive list of abbreviations would improve clarity.

(Answer) We added abbreviations below each figure and Table

Round 2

Reviewer 2 Report

The  Revisions ns are not adequately explained , and the manuscript needs to show all the changes highlited.

Author Response

The  Revisions ns are not adequately explained , and the manuscript needs to show all the changes highlited

(answer) We revised some words. Please check the points.

Round 3

Reviewer 2 Report

This   Revision is minimally sufficient. A more comprehensive revision would have been  better.